# ADVERSARIALLY ROBUST REPRESENTATIONS WITH SMOOTH ENCODERS

**A. Taylan Cemgil, Sumedh Ghaisas, Krishnamurthy Dvijotham & Pushmeet Kohli**
DeepMind, London
{taylancemgil,sumedhg,dvij,pushmeet}@google.com

## ABSTRACT

This paper studies the undesired phenomena of over-sensitivity of representations learned by deep networks to semantically-irrelevant changes in data. We identify a cause for this shortcoming in the classical Variational Auto-encoder (VAE) objective, the evidence lower bound (ELBO). We show that the ELBO fails to control the behaviour of the encoder out of the support of the empirical data distribution and this behaviour of the VAE can lead to extreme errors in the learned representation. This is a key hurdle in the effective use of representations for data-efficient learning and transfer. To address this problem, we propose to augment the data with specifications that enforce insensitivity of the representation with respect to families of transformations. To incorporate these specifications, we propose a regularization method that is based on a selection mechanism that creates a fictive data point by explicitly perturbing an observed true data point. For certain choices of parameters, our formulation naturally leads to the minimization of the entropy regularized Wasserstein distance between representations. We illustrate our approach on standard datasets and experimentally show that significant improvements in the downstream adversarial accuracy can be achieved by learning robust representations completely in an unsupervised manner, without a reference to a particular downstream task and without a costly supervised adversarial training procedure.

## 1 INTRODUCTION

Representation learning is a fundamental problem in Machine learning and holds the promise to enable data-efficient learning and transfer to new tasks. Researchers working in domains like Computer Vision (Krizhevsky et al., 2012) and Natural Language Processing (Devlin et al., 2018) have already demonstrated the effectiveness of representations and features computed by deep architectures for the solution of other tasks. A case in point is the example of the FC7 features from the AlexNet image classification architecture that have been used for many other vision problems (Krizhevsky et al., 2012).

The effectiveness of learned representations has given new impetus to research in representation learning, leading to a lot of work being done on the development of techniques for inducing representations from data having desirable properties like disentanglement and compactness (Burgess et al., 2018; Achille & Soatto, 2017; Bengio, 2013; Locatello et al., 2019). Many popular techniques for generating representation are based on the Variational AutoEncoders (VAE) model (Kingma & Welling, 2013; Rezende et al., 2014). The use of deep networks as universal function approximators has facilitated very rapid advancements which samples generated from these models often being indistinguishable from natural data. While the quality of generated examples can provide significant convincing evidence that a generative model is flexible enough to capture the variability in the data distribution, it is far from a formal guarantee that the representation is fit for other purposes. In fact, if the actual goal is learning good latent representations, evaluating generative models only based on reconstruction fidelity and subjective quality of typical samples is neither sufficient nor entirely necessary, and can be even misleading.

In this paper, we uncover the problematic failure mode where representations learned by VAEs exhibit over-sensitivity to semantically-irrelevant changes in data. One example of such problematic

behaviour can be seen in Figure 1. We identify a cause for this shortcoming in the classical Variational Auto-encoder (VAE) objective, the evidence lower bound (ELBO), that fails to control the behaviour of the encoder out of the support of the empirical data distribution. We show this behaviour of the VAE can lead to extreme errors in the recovered representation by the encoder and is a key hurdle in the effective use of representations for data-efficient learning and transfer. To address this problem, we propose to augment the data with properties that enforce insensitivity of the representation with respect to families of transformations.

To incorporate these specifications, we propose a regularization method that is based on a selection mechanism that creates a fictive data point by explicitly perturbing an observed true data point. For certain choices of parameters, our formulation naturally leads to the minimization of the entropy regularized Wasserstein distance between representations. We illustrate our approach on standard datasets and experimentally show that significant improvements in the downstream adversarial accuracy can be achieved by learning robust representations completely in an unsupervised manner, without a reference to a particular downstream task and without a costly supervised adversarial training procedure.

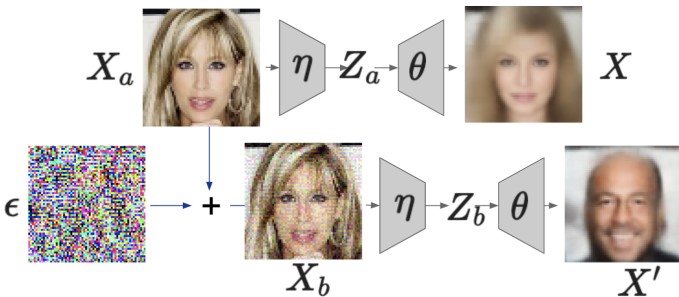

Figure 1: An illustration of the intrinsic fragility of VAE representations. Outputs from a Variational Autoencoder with encoder $f$ and decoder $g$ parametrized by $\eta$ and $\theta$, respectively, trained on CelebA. Conditioned on the encoder input $X_a = x_a$ the decoder output $X = g(f(x_a)) = (g \circ f)(x_a)$ is shown on the top row. When the original example is perturbed with a carefully selected vector $d$ such that $X_b = X_a + d$ with $\|d\| \leq \epsilon$, the output $X'$ turns out to be perceptually very different. Such examples suggest that either the representations $Z_a$ and $Z_b$ are very different (the encoder is not smooth), or the decoder is very sensitive to small changes in the representation (the decoder is not smooth), or both. We identify the source of the problem primarily as the encoder and propose a practical solution.

It is clear that if learned representations are overly sensitive to irrelevant changes in the input (for example, small changes in the pixels of an image or video, or inaudible frequencies added to an audio signal), models that rely on these representations are naturally susceptible to make incorrect predictions when inputs are changed. We argue that such specifications about the robustness properties of learned representations can be one of the tractable guiding features in the search for good representations. Based on these observations, we make the following contributions:

1. We introduce a method for learning robust latent representations by explicitly targeting a structured model that admits the original VAE model as a marginal. We also show that in the case the target is chosen a pairwise conditional random field with attractive potentials, this choice leads naturally to the Wasserstein divergence between posterior distributions over the latent space. This insight provides us a flexible class of robustness metrics for controlling representations learned by VAEs.

2. We develop a modification to training algorithms for VAEs to improve robustness of learned representations, using an external selection mechanism for obtaining transformed examples and by enforcing the corresponding representations to be close. As a particular selection mechanism, we adopt attacks in adversarial supervised learning (Madry et al., 2017) to attacks to the latent representation. Using this novel unsupervised training procedure we learn encoders with adjustable robustness properties and show that these are effective at learning representations that perform well across a variety of downstream tasks.

3. We show that alternative models proposed in the literature, in particular $\beta$-VAE model used for explicitly controlling the learned representations, or Wasserstein Generative Adversarial Networks (GANs) can also be interpreted in our framework as variational lower bound maximization.

4. We show empirically using simulation studies on MNIST, color MNIST and CelebA datasets, that models trained using our method learn representations that provide a higher degree of adversarial robustness even without supervised adversarial training.

## 2 GENERATIVE MODELS

Modern generative models are samplers $p(X|\theta)$ for generating realizations from an ideal target distribution $\pi(X)$, also known as the data distribution. In practice $\pi(X)$ is unknown in the sense that it is hard to formally specify. Instead, we have a representative data set $\mathcal{X}$, samples that are assumed to be conditionally independently drawn from the data distribution $\pi(X)$ of interest. We will refer to the empirical distribution as $\hat{\pi}(X) = \frac{1}{|\mathcal{X}|} \sum_{\xi \in \mathcal{X}} \delta(x - \xi)$. The goal is learning a parameter $\theta^*$ such that $p(X|\theta = \theta^*) = \int dZ p(X|Z, \theta = \theta^*) p(Z) \approx \hat{\pi}(X)$, thereby also learning a generator.

### 2.1 FROM VAE TO SMOOTH ENCODERS

The VAE corresponds to the latent variable model $p(X|Z, \theta)p(Z)$ with latent variable $Z$ and observation $X$. The forward model $p(X|Z = z, \theta)$ (the decoder) is represented using a neural network $g$ with parameters $\theta$, usually the mean of a Gaussian $\mathcal{N}(X; g(z; \theta), vI_x)$ where $v$ is a scalar observation noise variance and $I_x$ is an identity matrix. The prior is usually a standard Gaussian $p(Z = z) = \mathcal{N}(z; 0, I_z)$. The exact posterior over latent variables $p(Z|X = x, \theta)$ is approximated by a probability model $q(Z|X = x, \eta)$ with parameters $\eta$. A popular choice here is a multivariate Gaussian $\mathcal{N}(Z; \mu(x; \eta), \Sigma(x; \eta))$, where the mapping $f$ such that $(\mu, \Sigma) = f(x, \eta)$ is chosen to be a neural network (with parameters $\eta$ to be learned from data). We will refer to the pair $f, g$ as an encoder-decoder pair. Under the above assumptions, VAE's are trained by maximizing the following form of the ELBO using stochastic gradient descent (SGD),

$$\log p(X = x|\theta) \geq \mathbb{E}\left\{\log p(X = x|Z, \theta)\right\}_{q(Z|X=x,\eta)} - D_{\mathrm{KL}}(q(Z|X = x, \eta)||p(Z)) \equiv \mathcal{B}_x(\eta, \theta)$$

The gradient of the Kullback-Leibler (KL) divergence term above (see A.1) is available in closed form. An unbiased estimate of the gradient of the first term can be obtained via sampling $z$ from $q$ using the reparametrization trick Kingma & Welling (2013), aided by automatic differentiation.

### 2.2 A PROBLEM WITH THE VAE OBJECTIVE

Under the i.i.d. assumption, where each data point $x^{(n)}$, for $n = 1 \ldots N$ is independently drawn from the model an equivalent batch ELBO objective can be defined (See also E.1) as

$$\mathcal{B}(\eta, \theta) \equiv \frac{1}{N} \sum_{n=1}^{N} \mathcal{B}_{x^{(n)}}(\eta, \theta) = -D_{\mathrm{KL}}(\hat{\pi}(X)q(Z|X, \eta)||p(X|Z, \theta)p(Z)) + \text{const} \tag{1}$$

where the empirical distribution of observed data is denoted as $\hat{\pi}$. This form makes it more clear that the variational lower bound is only calculating the distance between the encoder and decoder under the support of the empirical distribution.

To see how this locality leads to a fragile representation, we construct a VAE with discrete latents and observations. We let $X \in \{1, \ldots, N_x\}$ and $Z \in \{1, \ldots, N_z\}$ and define the following system of conditional distributions as the decoder and encoder models as:

$$p(X = i|Z = j) \propto \exp\left(\frac{1}{v}\omega\left(m_j - i/N_x\right)\right) \quad q(Z = j|X = i) \propto \exp\left(\frac{1}{\sigma_i}\omega\left(\mu_i - j/N_z\right)\right)$$

where $\omega(u) = \cos(2\pi u)$. These distributions can be visualized by heatmaps of probability tables where $i$ and $j$ are row and column indicies, respectively Figure 2. This particular von-Mises like parametrization is chosen for avoiding boundary effects due to a finite latent and observable spaces.

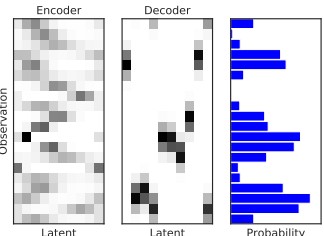

Figure 2: Example VAE model. (left) Heatmap of the encoder distribution (darker colors referring to higher probability) $q(Z = j|X = i; \mu_i, \sigma_i)$ where each row $i$ is a probability distribution over latents with a mode around $\mu_i$ and spread $\sigma_i$ (middle) Heatmap of the decoder distribution $p(X = i|Z = j, m_j, v_j)$ where each column $j$ is a probability distribution with mode at $m_j$ and spread $v$. The prior $p(Z)$ is chosen to be uniform and is not shown here. (right) The marginal model $p(X = i|m, v) = \sum_{j=1}^{N_z} p(Z = j)p(X = i|Z = j, m_j, v)$ depicted as an histogram.

The prior $p(Z)$ is taken as uniform, and is not shown. Note that this parametrization emulates a high capacity network that can model any functional relationship between latent states and observations, while being qualitatively similar to a standard VAE model with conditionally Gaussian decoder and encoder functions.

In reality, the true target density is not available but we would have a representative sample. To simulate this scenario, we sample a 'dataset' from a discrete target distribution $\pi(X)$: this is merely a draw from a multinomial distribution, yielding a multinomial vector $s$ with entries $s_i$ that gives the count how many times we observe $x = i$. The results of such an experiment are depicted in Figure 3(a) (see caption for details). This picture reveals several important properties of a VAE approximation.

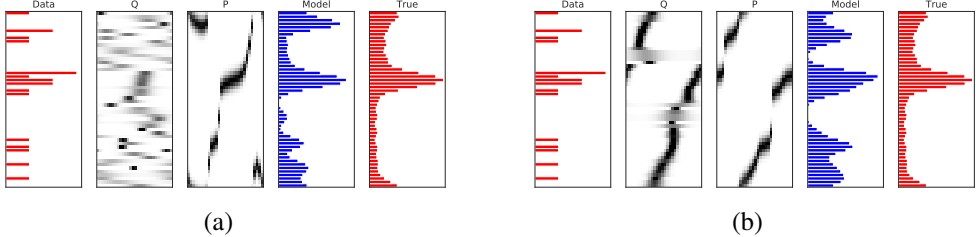

(a)                        (b)

Figure 3: (a) Result by optimizing the ELBO for a VAE that illustrates the fragility of the encoder. Subfigure with the title 'Data' ($\hat{\pi}(X)$) is a random sample from the true target 'Target' ($\pi(X)$) on the right. The resulting encoder $q(Z|X)$ and decoder $p(X|Z)$ are shown as 'Q' and 'P', respectively. The vertical and horizontal axes correspond to latents $Z$ and observations $X$ respectively. Integrating over the decoder distribution using a uniform prior $p(Z)$ over the latents, we obtain the model distribution 'Model' $p(X) = \sum_Z p(X|Z)p(Z)$. (b) The results obtained by a smooth encoder. Both the decoder and the representation (encoder) are more smooth while essentially having a similar fitting quality.

1. After training, we observe that when $j$ and $j'$ are close, the corresponding conditionals $p(X|Z = j)$ and $p(X|Z = j')$ are close (hence corresponding decoder mean parameters $m_j$ and $m_{j'}$ are close, hence (see middle panel of Fig.3(a) with the title $P$ showing the decoder). This smoothness is perhaps surprising at a first sight: in this example, we could arbitrarily permute columns of the decoder and still get the same marginal distribution. Technically speaking, given a uniform prior $p(Z)$, the marginal likelihood $p(X|\theta)$ is entirely invariant with respect to permutations of the latent state. In fact if the encoder distribution wouldn't be constrained we could also permute the columns of the encoder to keep the ELBO invariant. In the appendix E.2, we provide an argument why the choice of an unimodal encoder model and optimization of the variational objective leads naturally to smooth decoder functions.

2. The encoders found by the VAE on the other hand are not smooth at all, despite the fact that the model shows a relatively good fit. This behaviour alerts us about judging generative models only by the quality of the samples, by traversing the latent space and generating conditional samples from the decoder. The quality of the decoder seems to be not a proxy for the robustness of the representation.

The fragility of representations is inherent from the ELBO objective. For the entire dataset, a batch ELBO that involves the counts $s_i$ can be written as

$$\text{ELBO} \quad = \quad -\sum_i \sum_j s_i q(Z = j | X_a = i) \log \frac{s_i q(Z = j | X_a = i)}{p(X = i | Z = j) p(Z = j)} \tag{2}$$

The last expression is proportional to the negative KL divergence between two tabular distributions: $s_i q(Z = j | X_a = i)/L$ and $p(X = i | Z = j) p(Z = j)$. As such, whenever $s_i$ is zero, the contribution of row $i$ of the encoder distribution vanishes and the corresponding parameters $\mu_i$ and $\sigma_i$ are not effecting the lower bound. In a sense, the objective does not enforce any structure on the encoder outside of the position of the data points in the training set. This figure shows that the out-of-sample behaviour (i.e., for $i$ where $\hat{\pi}(X) = 0$) the encoder is entirely initialization dependent, hence no learning takes place. We would also expect that the resulting representations would be fragile, in the sense that a small perturbation of an observation can result in a large change in the encoder output.

## 3 ROBUST REPRESENTATIONS WITH SMOOTH ENCODERS

In this section, we will adopt a strategy for training the encoder that is guaranteed not to change the original objective of the decoder when maximizing the lower bound while obtaining a smoother representation. The key idea of our approach is that we assume an *external selection mechanism* that is able to provide new fictive data point $x'$ in the vicinity of each observation in our data set $x$. Here, "in the vicinity" means that we desire that the corresponding latent state of the original datapoint $z = f(x; \eta)$ and the latent state of the fictitious point $z' = f(x'; \eta)$ should be close to each other in some sense. Assuming the existence of such an external selection mechanism, we first define the following augmented distribution

$$p(X = x, X' = x' | \theta) \propto \int p(X = x | Z_a, \theta) p(X' = x' | Z_b, \theta) \psi(Z_a, Z_b) dZ_a dZ_b \tag{3}$$

where $\psi(Z_a, Z_b) = \exp(-\frac{\gamma}{2} c(Z_a, Z_b)) p(Z_a) p(Z_b)$. This is a pairwise conditional Markov random field (CRF) model (Lafferty et al., 2001; Sutton & McCallum, 2012), where we take $c(Z_a, Z_b)$ as a pairwise cost function. A natural choice here would be, for example, the Euclidean square distance $\|Z_a - Z_b\|^2$. Moreover, we choose a nonnegative coupling parameter $\gamma \geq 0$. For any pairwise $Q(Z_a, Z_b)$ distribution, the ELBO has the following form

$$
\begin{aligned}
\log p(X = x, X' = x' | \theta) \quad \geq \quad & \mathbb{E}\left\{\log p(X = x | Z_a, \theta)\right\}_{Q(Z_a)} + \mathbb{E}\left\{\log p(Z_a)\right\}_{Q(Z_a)} \\
& + \mathbb{E}\left\{\log p(X' = x' | Z_b, \theta)\right\}_{Q(Z_b)} + \mathbb{E}\left\{\log p(Z_b)\right\}_{Q(Z_b)} \\
& - \frac{\gamma}{2} \mathbb{E}\left\{c(Z_a, Z_b)\right\}_{Q(Z_a, Z_b)} + H(Q(Z_a, Z_b))
\end{aligned}
\tag{4}
$$

It may appear that the SE has to maintain a pairwise approximation distribution $Q(Z_a, Z_b)$. However, this turns out to be not necessary. Given the encoder, the marginals of $Q(Z_a, Z_b)$ are fixed as $Q_a(Z_a) = q(Z | X_a = x, \eta)$ and $Q_b(Z_b) = q(Z | X_b = x, \eta)$, so the only remaining terms that depend on the pair distribution are the final two terms in (4). We note that this two terms are just the objective function of the entropy regularized optimal transport problem (Cuturi, 2013; Amari et al., 2017). If we view $Q(Z_a, Z_b)$ as a transport plan, the first term is maximal when the expected cost is minimal while the second term is maximal when the variational distribution is factorized as $Q(Z_a, Z_b) = Q_a(Z_a) Q_b(Z_b)$.

In retrospection, this link is perhaps not that surprising as the Wasserstein distance, the solution of the optimal transport problem, is itself defined as the solution to a variational problem (Solomon, 2018): Consider a set $\Gamma$ of joint densities $Q(Z_a, Z_b)$ with the property that $Q$ has fixed marginals $Q_a(Z_a)$ and $Q_b(Z_b)$, i.e.,

$$\Gamma[Q_a, Q_b] \equiv \left\{ Q : Q_a(Z_a) = \int Q(Z_a, Z_b) dZ_b, \ Q_b(Z_b) = \int Q(Z_a, Z_b) dZ_a \right\} \tag{5}$$

The Wasserstein divergence[1], denoted by $\mathcal{WD}$ is defined as the solution of the optimization problem with respect to pairwise distribution $Q$

$$\mathcal{WD}[c](Q_a, Q_b) = \inf_{Q \in \Gamma} \int c(Z_a, Z_b) Q(Z_a, Z_b) dZ_a dZ_b \tag{6}$$

where $c(Z_a, Z_b)$ is a function that specifies the 'cost' of transferring a unit of probability mass from $Z_a$ to $Z_b$.

It is important to note that with our choice of the particular form of the variational distribution $Q(Z_a, Z_b)$ we can ensure that we are still optimizing a lower bound of the original problem. We can achieve this by simply integrating out the $X'$, effectively ignoring the likelihood term for the fictive observations. Our choice does not modify the original objective of the decoder due to the fact that the marginals are fixed given $\eta$. To see this, take the exponent of (4) and integrate over the unobserved $X'$

$$
\begin{aligned}
\log p(X = x | \theta) &= \log \int dX' p(X = x, X' | \theta) \\
&\geq \mathbb{E} \left\{ \log p(X = x | Z_a, \theta) \right\}_{Q(Z_a)} + \mathbb{E} \left\{ \log p(Z_a) \right\}_{Q(Z_a)} + \mathbb{E} \left\{ \log p(Z_b) \right\}_{Q(Z_b)} \\
&\quad - \frac{\gamma}{2} \mathbb{E} \left\{ c(Z_a, Z_b) \right\}_{Q(Z_a, Z_b)} + H(Q(Z_a, Z_b)) \equiv \mathcal{B}_{\text{SE}}(\theta, \eta)
\end{aligned} \tag{7}
$$

we name this lower bound $\mathcal{B}_{\text{SE}}$ as the Smooth Encoder ELBO (SE-ELBO). The gradient of $\mathcal{B}_{\text{SE}}$ with respect to the decoder parameters $\theta$ is identical to the gradient of the original VAE objective $\mathcal{B}$. This is intuitive as $x'$ is an artificially generated sample, we should use only terms that depend on $x$ and not on $x'$. Another advantage of this choice is that it is possible to optimize the decoder and encoder concurrently as in the standard VAE. Only an additional term enters for the regularization of the encoder where the marginals obtained via amortized inference $q(Z_a | x_a, \eta)$ and $q(Z_b | x_b, \eta)$ are forced to be close in a regularized Wasserstein distance sense, with the coupling strength $\gamma$. Effectively, we are doing data augmentation for smoothing the representations obtained by the encoder without changing the actual data distribution. In the appendix E.3, we also provide an argument about the smoothness of the corresponding encoder mapping, justifying the name. The resulting algorithm is actually a simple modification to the standard VAE and is summarized below:

Initialize $\eta^{(0)}, \theta^{(0)}$
**for** $\tau = 1, 2, \dots$ **do**
    $x_a = \text{GetData}(), \; x_b = \text{Select}(x_a; L, \epsilon)$ (see Section 3.1)
    $\mu_a, \Sigma_a = f(x_a; \eta), \qquad \mu_b, \Sigma_b = f(x_b; \eta)$ (Compute Representation)
    $\mathcal{WD}_{2,\gamma}(\eta) = \text{EntropyRegularizedWassersteinDivergence}(\mu_a, \Sigma_a, \mu_b, \Sigma_b, \gamma)$ (see Apdx. B.2)
    $u \sim \mathcal{N}(0, I)$ (Reparametrization Trick)
    $E_1(\eta, \theta) = -\frac{1}{2v} \| x_a - g(\mu_a + \Sigma_a^{1/2} u; \theta) \|^2$ (Data Fidelity)
    $E_2(\eta) = -\frac{1}{2} \left( \| \mu_a \|^2 + \| \mu_b \|^2 + \mathbf{Tr} \{ \Sigma_a + \Sigma_b \} \right)$ (Prior Fidelity)
    $E(\eta, \theta) = E_1(\eta, \theta) + E_2(\eta) - \mathcal{WD}_{2,\gamma}(\eta)$
    $\eta^{(\tau)}, \theta^{(\tau)} = \text{Optimization Step}(E, \eta^{(\tau-1)}, \theta^{(\tau-1)})$
**end**

### 3.1 SELECTION MECHANISM VIA ADVERSARIAL ATTACKS

Adversarial attacks are one of the most popular approaches for probing trained models in supervised tasks, where the goal of an adversarial attack is finding small perturbations to an input example that would maximally change the output, e.g., flip a classification decision, change significantly a prediction (Szegedy et al., 2013). The perturbed input is named as an adversarial example and these extra examples are used, along with the original data points, for training adversarially robust models (Madry et al., 2017; Kurakin et al., 2016). As extra samples are also included, such a training procedure is referred as *data augmentation*. However, in unsupervised learning and density estimation, data augmentation is not a valid approach as the underlying empirical distribution would be altered by the introducing new points.

---

[1]We use the term divergence to distinguish the optimal transport cost from the corresponding metric. This distinction is reminiscent to the distinction between Euclidian divergence $\| \cdot \|^2$ and the Euclidian distance $\| \cdot \|$

However, as we let the encoder to target a different distribution than the actual decoder, we can actually use the extra, self generated samples to improve desirable properties of a given model. Hence this approach could also be interpreted as a 'self-supervised' learning approach where we bias our search for a 'good encoder' and the data selection mechanism acts like a critique, carefully providing examples that should lead to similar representations.

In this paper we will restrict ourselves to Projected Gradient Descent (PGD) attacks popular in adversarial training Carlini & Wagner (2016) as a selection mechanism, where the goal of the attacker is finding a point that would introduce the maximum difference in the Wasserstein distance of the latent representation. In other words, we implement our selection mechanism where the extra data point is found by approximately solving the following constrained optimization problem

$$x' = x + \arg \max_{d:\|d\|_p \leq \epsilon} \mathcal{WD}(q(Z|X = x, \eta), q(Z|X' = x + d, \eta))$$

This attack is assigned a certain iteration budget $L$ for a given radius $\epsilon$, that we refer as *selection iteration budget* and the *selection radius*, respectively. We note a similar attack mechanism is proposed for generative models as described in (Kos et al., 2017), where one of the proposed attacks is directly optimizing against differences in source and target latent representations. Note that our method is not restricted to a particular selection mechanism; indeed two inputs that should give a similar latent representation could be used as candidates.

## 4 EXPERIMENTS

**Goal and Protocol** In our experiments, we have tested and compared the adversarial accuracy of representations learned using a VAE and our smooth encoder approach. We adopt a two step experimental protocol, where we first train encoder-decoder pairs agnostic to any downstream task. Then we fix the representation, that is we freeze the encoder parameters and only use the mean of the encoder as the representation, then train a simple linear classifier based on the fixed representation using standard techniques. In this supervised stage, no adversarial training technique is employed. Ideally, we hope that such an approach will provide a degree of adversarial robustness, without the need for a costly, task specific adversarial training procedure. To evaluate the robustness of the resulting classifier, for each data point in the test set, we search for an adversarial example using an untargeted attack that tries to change the classification decision. The adversarial accuracy is reported in terms of percentage of examples where the attack is not able to find an adversarial example.

The VAE and SE decoder and encoder are implemented using standard MLP and ConvNet architectures. The selection procedure for SE training is implemented as a projected gradient descent optimization (a PGD attack) with selection iteration budget of $L$ iterations to maximize the Wasserstein distance between $q(Z|X = x)$ and $q(Z|X = x + \delta)$ with respect to the perturbation $\delta$ where $\|\delta\|_\infty < \epsilon$. Further details about the experiment can be found in the appendix C.1.

**Results:** We run simulations on ColorMNIST, MNIST and CelebA datasets. The ColorMNIST is constructed from the MNIST dataset by coloring each digit artificially with all of the colors corresponding to the seven of the eight corners of the RGB cube (excluding black). We present the results with the strongest attack we have experimented: a PGD attack with 100 iterations and 10 restarts. We observe that for weaker attacks (such as 50 iterations with no restarts), the adversarial accuracy is typically much higher.

For the ColorMNIST dataset, the results are shown in Figure 4 where we test the adversarial accuracy of representations learned by our method and compare it to a VAE. We observe that the adversarial accuracy of a VAE representation quickly drops towards zero while SE can maintain adversarial accuracy in both tasks. In particular, we observe that for the arguably simpler color classification task, we are able to obtain close to perfect adversarial test accuracy using representations learned by the VAE and SE. However, when the classifiers are attacked using PGD, the adversarial accuracy quickly drops with increasing radius size, while the accuracy degrades more gracefully in the SE case.

In Figure 5, we show the robustness behaviour of the method for different architectures. A ConvNet seems to perform relatively better than an MLP but these results show that the VAE representation is not robust, irrespective of the architecture. We have also carried out controlled experiments with random selection instead of the more costly untargeted adversarial attacks (See appendix C.1 Fig-

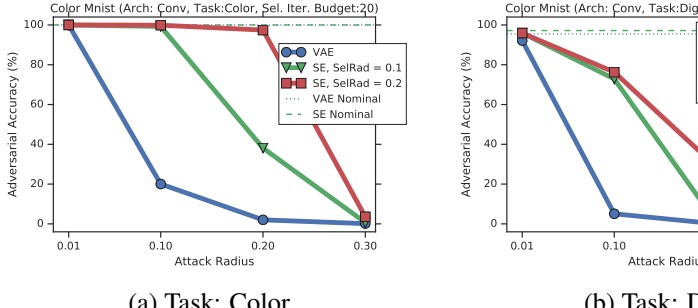

(a) Task: Color                    (b) Task: Digit

Figure 4: Simulation Results on ColorMNIST. The goal is the comparison of adversarial accuracy of VAE representations with SE representations trained with a selection radius of $0.1$ and $0.2$ and a selection budget of 20 PGD iterations. Vertical axis shows the adversarial accuracy as a function of attack radius. The dashed and dotted lines show the nominal accuracy of the VAE and SE when there are no attacks (The SE nominal accuracy is virtually identical for different selection radii, hence only a single level is shown.)

ure 7(a) for further results). We observe some limited improvements with SE using random selection in adversarial accuracy compared to VAE but training a SE with adversarial selection seems to be much more effective. We note that the selection iteration budget was lower ($L = 20$ with no restarts) than the attack iteration budget (100 with 10 restarts) during evaluation. It was not practical to train the encoder with more powerful selection attacks, thus it remains to be seen if the tendency of increased accuracy with increased iteration budgets would continue. We also observe that essentially the same level of adversarial accuracy can be obtained with a small fraction of the available labels (See appendix C.1 Figure 8 for further results).

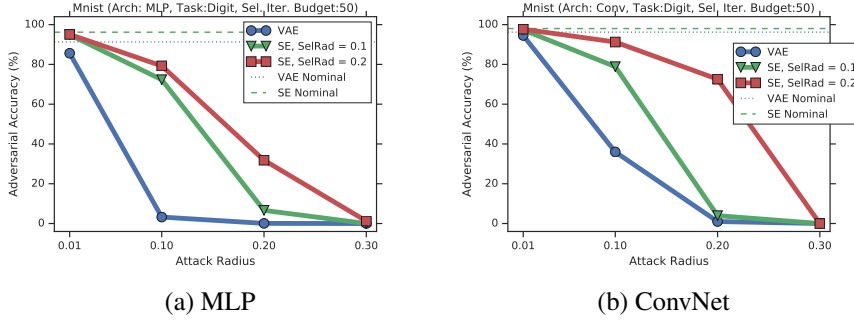

(a) MLP                       (b) ConvNet

Figure 5: Simulation Results on MNIST. The goal is illustrating the effect of the architecture (MLP and ConvNet). In all the examples, the SE is trained by a selection radius a budget of $50$ PGD iterations. The linear classifier is always trained without any adversarial training, by fixing the encoder parameters. The blue dot, green triangle and red squares correspond to the standard VAE and SE trained with a selection radius of $0.1$ and $0.2$ respectively. The dashed and dotted lines show the nominal accuracy of the VAE and SE when there are no attacks (The SE nominal accuracy is virtually identical for different selection radii, hence only a single level is shown.)

We have also repeated our experiments on the CelebA dataset, a large collection of high resolution face images labeled with $40$ attribute labels per example. We have used $17$ of the attribute labels as the targets of $17$ different downstream classification tasks. The results are shown in Table.2. The results clearly illustrate that we can achieve much more robust representations than a VAE. It is also informative to investigate specific adversarial examples to understand the failure modes. In Figure 6 we show two illustrative examples from the CelebA. Here we observe that attacks to the SE representations are much more structured and semantically interpretable. In our exploratory investigations, we qualitatively observe that the reconstructions corresponding to the adversarial examples are almost always legitimate face images with clearly recognizable features. This also seems to support our earlier observation that VAE decoders are typically smooth while the encoders

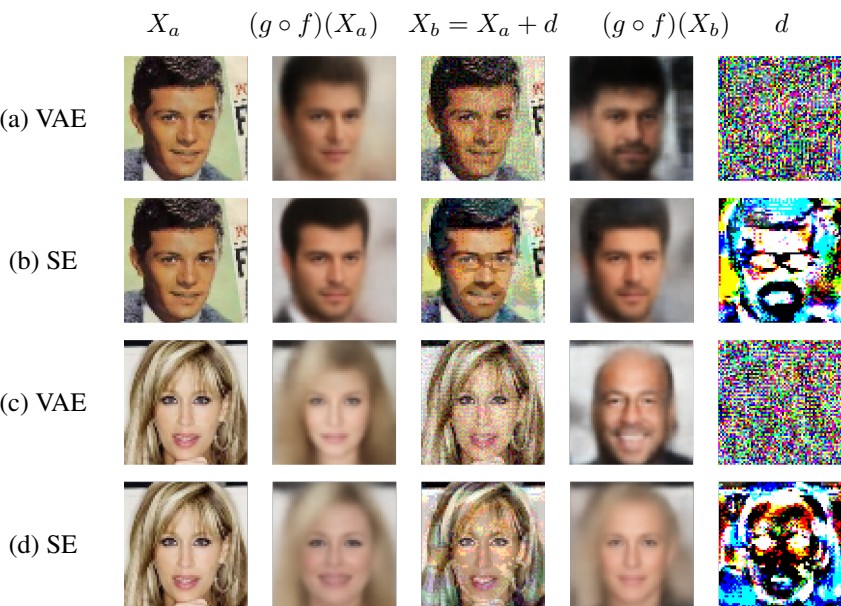

$$X_a \qquad (g \circ f)(X_a) \qquad X_b = X_a + d \qquad (g \circ f)(X_b) \qquad d$$

(a) VAE

(b) SE

(c) VAE

(d) SE

Figure 6: Qualitative results on CelebA. Attacks to downstream tasks (a), (b) 'Mustache' classification, (c), (d) 'Bald' classification. (a) In the VAE case, the attack is successful but the perturbation does not have a visible structure. (b) The SE representation is attacked by a perturbation that can clearly be identified as drawing a beard on the image. In this case, the attack is able to fool the classifier and the generated image from the representation is that of a person with beard and mustache. In the second example (c), the VAE representation seems to be attacked by exploiting the non-smooth nature of the encoder by mapping the latent representation to the one in the vicinity of a clearly different person with the desired features, as can be seen from the corresponding reconstruction. In contrast, in the SE case (d), an unsuccessful attack that adds a much more structured perturbation. From the reconstruction it is evident that a latent feature is attacked that seems to control the receding hairline.

are inferring non-robust features. Our approach seems to be a step towards obtaining more robust representations.

| Task | VAE Nom | VAE Adv | SE Nom | SE Adv | Task | VAE Nom | VAE Adv | SE Nom | SE Adv |
|---|---|---|---|---|---|---|---|---|---|
| Bald | 97.8 | 0.0 | 97.4 | 86.5 | Brown Hair | 83.1 | 0.0 | 80.5 | 41.5 |
| Mustache | 96.1 | 0.0 | 95.7 | 84.4 | Eyeglasses | 95.6 | 0.0 | 95.7 | 33.0 |
| Necklace | 86.2 | 0.0 | 88.0 | 78.9 | Black Hair | 79.7 | 0.0 | 81.4 | 31.4 |
| Straight Hair | 79.1 | 0.0 | 78.7 | 77.3 | Bangs | 90.3 | 0.0 | 89.6 | 27.0 |
| Wearing Hat | 96.6 | 0.0 | 96.4 | 77.3 | No Beard | 86.5 | 0.0 | 85.3 | 24.3 |
| Earrings | 79.3 | 0.0 | 81.3 | 55.3 | Wavy Hair | 71.4 | 0.0 | 72.8 | 10.2 |
| Blond Hair | 92.4 | 0.0 | 90.7 | 53.5 | Smiling | 82.7 | 0.0 | 85.7 | 1.1 |
| Necktie | 93.2 | 0.0 | 92.7 | 51.7 | Gender | 81.1 | 0.0 | 81.6 | 0.7 |
| | | | | | Lipstick | 79.2 | 0.0 | 80.3 | 0.6 |

Table 1: Comparison of nominal (Nom) and adversarial (Adv) accuracy (in percentage) on 17 downstream tasks using a VAE and a SE trained with a selection radius of $\epsilon = 0.1$ and evaluated with attack radius of 0.1 and iteration budget of 100 with 10 restarts. See Section 4 for the experimental protocol.

## 5 RELATED WORK

The literature on deep generative models and representation learning is quite extensive and is rapidly expanding. There is a plethora of models, but some approaches have been quite popular in recent

years: Generative Adversarial Networks (GANs) and VAEs. While the connection of our approach to VAE's is evident, there is also a connection to GANs. In the appendix, we provide the details where we show that a GAN decoder can be viewed as an instance of a particular smooth encoder. Our method is closely related to the $\beta$-VAE (Higgins et al., 2017), used for controlling representations replaces the original variational objective (1) with another one for explicitly trading the data fidelity with that of prior fidelity. In the appendix, we show that the method can be viewed as an instance of the smooth encoders.

Wasserstein distance minimization has been applied in generative models as an alternative objective for fitting the decoder. Following the general framework sketched in Bousquet et al. (2017), the terms of the variational decomposition of the marginal likelihood can be modified in order to change the observation model or the regulariser. For example, Wasserstein AutoEncoders (WAE) Tolstikhin et al. (2017), Zhang et al. (2019) or sliced Wasserstein Autoencoders Kolouri et al. (2018) propose to replace data fidelity and/or the KL terms with a Wasserstein distance. Our approach is different from these approaches as we do not propose to replace the likelihood as a fundamental principle for data fitting. In contrast, the Wasserstein distance formulation naturally emerges from the particular model choice and the corresponding variational approximation.

Our approach involves an adversarial selection step. The word 'Adversarial' is an overloaded term in generative modelling so it is important to mention differences between our approach. Adversarial Variational Bayes is a well known technique in the literature that aims to combine the empirical success of GANs with the probabilistic formulation of VAEs, where the limiting functional form of the variational distribution can be replaced by blackbox inference (Mescheder et al., 2017). This approach also does not modify the original VAE objective, however, the motivation here is different as the aim is developing a more richer family. In our view, for learning useful representations, when the decoder is unknown, the advantage of having a more powerful approximating family is not clear yet; this can even make the task of learning a good representation harder. Adversarial Autoencoders (Makhzani et al., 2015), Adversarially Learned Inference (ALI) (Dumoulin et al., 2016) and BiGANs (Bidirectional GANs) (Donahue et al., 2016) are also techniques that combine ideas from GANs and VAEs for learning generative models. The key idea is matching an encoder process $q(z|x)p(x)$ and to the decoder process $p(z)p(x|z)$ using an alternative objective, rather than by minimizing the KL divergence. In this formulation, $p(x)$ is approximated by the empirical data distribution, and $p(z)$ is the prior model of a VAE. The encoder $q(z|x)$ and decoder $p(x|z)$ are modelled using deep networks. This approach is similar to Wasserstein autoencoders that propose to replace the likelihood principle.

The idea of improving VAEs by capturing the correlation structure between data points using MRFs and graphical models has been also been recently proposed (Tang et al., 2019) under the name Correlated Variational Auto-Encoders (CVAEs). Our approach is similar, however we introduce the correlation structure not between individual data points but only between true data points and artificially selected data points. We believe that correctly selecting such a correlation structure of the individual data points can be quite hard in practice, but if such prior knowledge is available, CVAE can be indeed a much more powerful model than a VAE. We note that a proposal for automatically learning such a correlation structure is also recently proposed by (Louizos et al., 2019).

## 6 DISCUSSION AND CONCLUSIONS

In this paper, we have introduced a method for improving robustness of latent representations learned by a VAE. It must be stressed that our goal is not building the most powerful adversarially robust supervised classifier, but obtaining a method for learning generic representations that can be used for several tasks; the tasks can be even unknown at the time of learning the representations. While the nominal accuracy of an unsupervised approach is expected to be inferior to a supervised training method that is informed by extra label information, we observe that significant improvements in adversarial robustness can be achieved by our approach that forces smooth representations.

### ACKNOWLEDGMENTS

We are grateful to Arnaud Doucet for the insights and references about optimal transport and Arnaud Doucet, Johannes Welbl, Sven Gowal and Michalis Titsias for their helpful comments to earlier drafts of this paper.

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

# A  APPENDIX

## A.1  KL DIVERGENCE

The KL divergence between two Gaussian distributions translates to a well known divergence in the parameters (in the general case this is a Bregman divergence)

$$KL(P_a||P_b) = \frac{1}{2}\left(\mathbf{Tr}\,\Sigma_b^{-1}(\Sigma_a - \Sigma_b) - \log|\Sigma_b^{-1}\Sigma_a|\right) + \frac{1}{2}(\mu_a - \mu_b)^\top \Sigma_b^{-1}(\mu_a - \mu_b) \tag{8}$$

where $P_a = \mathcal{N}(\mu_a, \Sigma_a)$ and $P_b = \mathcal{N}(\mu_b, \Sigma_b)$ are Gaussian densities with mean $\mu.$ and covariance matrix $\Sigma.$, and $|\cdot|$ denotes the determinant for a matrix argument, and $\mathbf{Tr}$ denotes the trace. The KL divergence consists of two terms, the first term is the scale invariant divergence between two covariance matrices also known as a Itakuro-Saito divergence and the second term is a Mahalonobis distance between the means. The KL divergence is invariant to the choice of parametrization or the choice of the coordinate system.

# B  OPTIMAL TRANSPORT AND WASSERSTEIN DISTANCE

Consider a set $\Gamma$ of joint densities $Q(Z_a, Z_b)$ with the property that $Q$ has fixed marginals $Q_a(Z_a)$ and $Q_b(Z_b)$, i.e.,

$$\Gamma[Q_a, Q_b] \equiv \left\{ Q : Q_a(Z_a) = \int Q(Z_a, Z_b)dZ_b, \ Q_b(Z_b) = \int Q(Z_a, Z_b)dZ_a \right\} \tag{9}$$

The Wasserstein divergence $\mathcal{WD}$ is defined as the solution of the optimization problem with respect to pairwise distribution $Q$

$$\mathcal{WD}[c](Q_a, Q_b) = \inf_{Q \in \Gamma} \int c(Z_a, Z_b)Q(Z_a, Z_b)dZ_adZ_b \tag{10}$$

where $c(z_a, z_b)$ is a function that specifies the 'cost' of transferring a unit of probability mass from $z_a$ to $z_b$.

## B.1  $\ell_2$-WASSERSTEIN DISTANCE $\mathcal{W}$

The $\ell_2$-Wasserstein distance $\mathcal{W}_2^2$ for two Gaussians has an interesting form. The optimum transport plan, where the minimum of (10) is attained, is given

$$Q^*(z_a, z_b) = \mathcal{N}\left( \begin{pmatrix} \mu_a \\ \mu_b \end{pmatrix}, \begin{pmatrix} \Sigma_a & \Psi \\ \Psi & \Sigma_b \end{pmatrix} \right) \tag{11}$$

where $\Psi = \Sigma_a \Sigma_b^{1/2}(\Sigma_b^{1/2}\Sigma_a\Sigma_b^{1/2})^{-1/2}\Sigma_b^{1/2}$. It can be checked that this optimal Guassian density is degenerate in the sense that there exists a linear mapping between $z_a$ and $z_b$:

$$z_a(z_b) = \mu_a + \Sigma_a\Sigma_b^{1/2}(\Sigma_b^{1/2}\Sigma_a\Sigma_b^{1/2})^{-1/2}\Sigma_b^{-1/2}(z_b - \mu_b)$$

where $A^{1/2}$ denotes the matrix square root, a symmetric matrix such that $(A^{1/2})^2 = A$ for a symmetric positive semidefinite matrix $A$. The $\ell_2$-Wasserstein distance is the value attained by the optimum transport plan

$$\mathcal{W}_2^2(P_a, P_b) = \|\mu_a - \mu_b\|_2^2 + \mathbf{Tr}\left(\Sigma_a + \Sigma_b - 2\left(\Sigma_b^{1/2}\Sigma_a\Sigma_b^{1/2}\right)^{1/2}\right) \tag{12}$$

## B.2  ENTROPY REGULARIZED $\ell_2$-WASSERSTEIN DISTANCE

Entropy Regularized $\ell_2$-Wasserstein is the value attained by the minimizer of the following functional

$$F[Q] = \frac{\gamma}{2}\mathbb{E}\left\{\mathbf{Tr}(Z_a - Z_b)(Z_a - Z_b)^\top\right\}_{Q(Z_a, Z_b)} - H[Q(Z_a, Z_b)] \tag{13}$$

where $H$ is the entropy of the joint distribution $Q$. Using the form in (11) subject to the semidefinite constraint $\Sigma_a - \Psi\Sigma_b^{-1}\Psi^\top \succeq 0$

$$\mathbf{Tr}\left(z_a - z_b\right)\left(z_a - z_b\right)^\top = -2\,\mathbf{Tr}(\Psi) + \text{const} \tag{14}$$

The entropy of a Gaussian $Q(z_a, z_b)$ is given by the Schur formula

$$H[Q(z_a, z_b)] = \frac{D}{2}\log(2\pi e) + \frac{1}{2}\log|\Sigma_b||\Sigma_a - \Psi\Sigma_b^{-1}\Psi^\top| \tag{15}$$

Here, $D$ is the dimension of the vector $(z_a, z_b)$. The entropy regularized problem has a solution where we need to minimize

$$\tilde{F}(\Psi) = -\gamma\,\mathbf{Tr}(\Psi) - \frac{1}{2}\,\mathbf{Tr}\log\left|\Sigma_a - \Psi\Sigma_b^{-1}\Psi^\top\right| \tag{16}$$

Taking the derivative and setting to zero

$$\frac{\partial\tilde{F}(\Psi)}{\partial\Psi} = -\gamma I + \Sigma_b^{-1}\Psi^\top\left(\Sigma_a - \Psi\Sigma_b^{-1}\Psi^\top\right)^{-1} \tag{17}$$

we obtain a particular Matrix Ricatti equation

$$0 = -\Psi\Sigma_b^{-1}\Psi^\top - \frac{1}{\gamma}\Sigma_b^{-1}\Psi^\top + \Sigma_a \tag{18}$$

that gives us a closed form formula for the specific entropy regularized Wasserstein distance

$$\mathcal{W}_{2,\gamma}^2(\mathcal{N}(m_a, \Sigma_a), \mathcal{N}(m_b, \Sigma_b)) = \|m_a - m_b\|_2^2 + \mathbf{Tr}\{\Sigma_a + \Sigma_b - 2\Psi\} \tag{19}$$

$$\mathcal{WD}_{2,\gamma}(\mathcal{N}(m_a, \Sigma_a), \mathcal{N}(m_b, \Sigma_b)) \equiv \frac{\gamma}{2}\mathcal{W}_{2,\gamma}^2(\mathcal{N}(m_a, \Sigma_a), \mathcal{N}(m_b, \Sigma_b)) \tag{20}$$

$$- \frac{D}{2}\log(2\pi e) - \frac{1}{2}\log|\Sigma_b||\Sigma_a - \Psi\Sigma_b^{-1}\Psi^\top| \tag{21}$$

For the case of two univariate Gaussians, i.e., when the joint distribution has the form

$$Q(Z_a, Z_b) = \mathcal{N}\left(\begin{pmatrix} m_a \\ m_b \end{pmatrix}, \begin{pmatrix} \Sigma_a & \psi \\ \psi & \Sigma_b \end{pmatrix}\right)$$

the solution is given by the solution of the scalar quadratic equation.

$$f(\psi)' = \psi^2 + \frac{1}{\gamma}\psi - \Sigma_a\Sigma_b = 0 \tag{22}$$

$$\psi = -\frac{1}{2\gamma} \pm \frac{1}{2|\gamma|}\left(1 + 4\gamma^2\Sigma_a\Sigma_b\right)^{1/2} \tag{23}$$

We take the root that gives a feasible solution as the minimizer. In the scalar case, this is the solution that satisfies $\Sigma_a - \psi^2/\Sigma_b \geq 0$, or equivalently $\Sigma_a\Sigma_b \geq \psi^2$

$$\psi = \frac{1}{2\gamma}(u_\gamma(\Sigma_a, \Sigma_b) - 1) \tag{24}$$

where we have defined

$$u_\gamma(\Sigma_a, \Sigma_b) = \left(1 + 4\gamma^2\Sigma_b\Sigma_a\right)^{1/2}$$

It can be easily checked that the other root is infeasible. For the scalar $\psi$ case we obtain

$$\mathcal{WD}_{2,\gamma}(\mathcal{N}(m_a, \Sigma_a), \mathcal{N}(m_b, \Sigma_b)) = \frac{\gamma}{2}\left(\|m_a - m_b\|_2^2 + \Sigma_a + \Sigma_b\right) - \frac{1}{2}(u_\gamma(\Sigma_a, \Sigma_b) - 1)$$

$$+ \frac{1}{2}\log(u_\gamma(\Sigma_a, \Sigma_b) + 1) - \frac{1}{2}\log(2\Sigma_b\Sigma_a) - \log(2\pi) - 1$$

## C SUMMARY OF THE SMOOTH ENCODER ALGORITHM WITH FACTORIZED GAUSSIAN

Assume a factorized encoder distribution of form $q(Z_a|x, \eta) = \prod_{k=1}^{D_z}\mathcal{N}(Z_a^k; \mu_a^k, \Sigma_a^k)$ and $q(Z_b|x', \eta) = \prod_{k=1}^{D_z}\mathcal{N}(Z_b^k; \mu_b^k, \Sigma_b^k)$ where $D_z$ is the dimension of the latent representation, and

$\mu^k$ and $\Sigma^k$ are the $k$'th component of the output of a neural network with parameters $\eta$. Similarly, $x^i$ denotes the $i$'th component of the observation vector $x$ of size $D_x$. For optimization, we need an unbiased estimate of the gradient of the SE-ELBO with respect to encoder parameters $\eta$ and decoder parameters $\theta$:

$$
\begin{aligned}
\mathcal{B}_{SE}(\eta, \theta) = & \ \mathbb{E}\left\{\log p(X = x | Z_a, \theta)\right\}_{q(Z_a | x_a, \eta)} + \mathbb{E}\left\{\log p(Z_a)\right\}_{q(Z_a | x_a, \eta)} + \mathbb{E}\left\{\log p(Z_b)\right\}_{q(Z_b | x_b, \eta)} \\
& -\frac{\gamma}{2}\mathbb{E}\left\{c(Z_a, Z_b)\right\}_{q(Z_a, Z_b | x_a, x_b, \eta)} + H(q(Z_a, Z_b | x_a, x_b, \eta))
\end{aligned}
$$

Given $x$, we first select a fictive sample $x'$ via a selection mechanism, in this case as an adversarial attack as explained in section 3.1.

Sample a latent representation and calculate the associated prediction

$$
z_a \ \sim \ q(Z_a | X_a = x, \eta) = \mathcal{N}(Z_a; \mu_a, \Sigma_a) \qquad \bar{x} = g(z_a; \eta)
$$

The terms of the SE-ELBO can be calculated as

$$
\begin{aligned}
\mathbb{E}\left\{\log p(x | Z_a, \theta)\right\}_{q(Z_a | X_a = x, \eta)} &\approx -\frac{D_x}{2}\log 2\pi v - \frac{1}{2v}\sum_{i=1}^{D_x}(x_i - \bar{x}_i)^2 \\
\mathbb{E}\left\{\log p(Z_a)\right\}_{q(Z_a | X_a = x, \eta)} &= -\frac{1}{2}\sum_{k=1}^{D_z}((\mu_a^k)^2 + \Sigma_a^k) - \frac{D_z}{2}\log 2\pi \\
\mathbb{E}\left\{\log p(Z_b)\right\}_{q(Z_b | X_b = x', \eta)} &= -\frac{1}{2}\sum_{k=1}^{D_z}((\mu_b^k)^2 + \Sigma_b^k) - \frac{D_z}{2}\log 2\pi \\
\mathcal{WD}_{2,\gamma} &= \frac{\gamma}{2}\mathbb{E}\left\{\|Z_a - Z_b\|^2\right\}_{q(Z_a, Z_b | x_a, x_b, \eta)} - H(q(Z_a, Z_b | x_a, x_b, \eta))
\end{aligned}
$$

where

$$
\begin{aligned}
u_\gamma(\Sigma_a, \Sigma_b) &= \sqrt{1 + 4\gamma^2\Sigma_b\Sigma_a} \\
\mathcal{WD}_{2,\gamma} &= \frac{\gamma}{2}\sum_{k=1}^{D_z}\left(\|\mu_a^k - \mu_a^k\|_2^2 + \Sigma_a^k + \Sigma_b^k\right) \\
&\quad -\frac{1}{2}\sum_{k=1}^{D_z}\left((u_\gamma(\Sigma_a^k, \Sigma_b^k) - 1) - \log(u_\gamma(\Sigma_a^k, \Sigma_b^k) + 1) + \log(2\Sigma_b^k\Sigma_a^k)\right) \\
&\quad -D_z\log(2\pi) - D_z
\end{aligned}
$$

## C.1 Experimental Details and Further Results

We always train decoder-encoder pairs with identical architectures using both the standard VAE ELBO and the SE ELBO with a fixed $\gamma$. Then, in each case by fixing the encoder (that is essentially using the same representation) and by only using the mean activations of the encoders, we train linear classifiers using standard training for solving several downstream tasks.

For both encoder and decoder networks we use a 4 layer multi layer perceptron (MLP) and a convolutional network (ConvNET) architectures with 200 units of ReLU activations at each layer. We carried out experiments with latent space dimensions of $32, 64$ and $128$, corresponding to an output sizes of an encoder with $64, 128$ and $256$ units, with two units per dimensions to encode the mean and the log-variance parameters of a fully factorized Gaussian condition distribution. The training is done using the Adam optimizer. Each network (both the encoder and decoder) are randomly initialized and trained for 300K iterations.

| Tasks | MNIST | Digit |
|---|---|---|
| | Color-MNIST | Color, Digit |
| | CelebA | see table 1 |
| Training | Representation Dimension, $\dim(Z)$ | 32, 64, 128, 256 |
| VAE or SE | Observation noise variance, $v$ | 0.25, 0.5, 1., 2. |
| | Architecture | MLP, ConvNET |
| Training | Coupling Strength $\gamma$ | 0.01, 0.1, 1, 5, 10, 50 |
| SE Only | Selection PGD Radius $\epsilon$ | 0.01, 0.1, 0.2, 0.3 |
| | Selection PGD Iteration Budget $L$ | 1, 5, 10, 20, 50 |
| Evaluation | Attack PGD Radius to downstream task | 0.01, 0.1, 0.2, 0.3 |
| | Attack PGD Iteration Budget | 100 (10 random restarts) |

Table 2: Experiment Hyperparameters

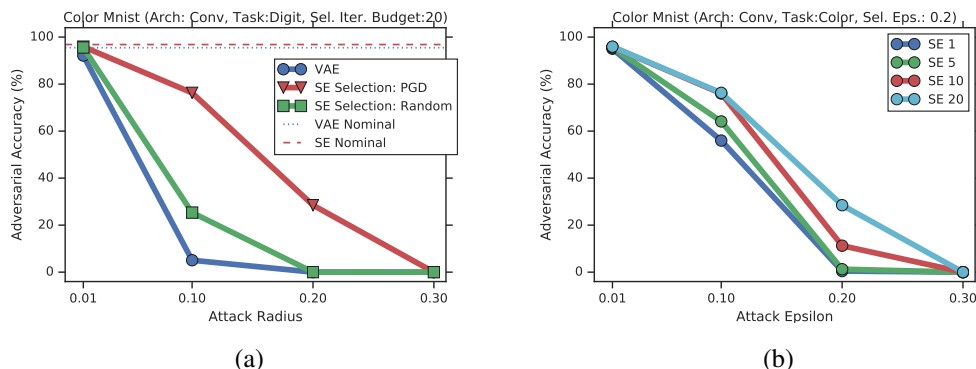

(a)                                     (b)

Figure 7: Simulation Results on ColorMNIST. (a) The goal is comparing the robustness as a result of the selection procedures, uniformly random selection from the unit ball and adversarial selection. (b) The effect of the selection budget $L$ on the adversarial accuracy. The plot shows adversarial accuracy results for $L = 1, 5, 10, 20$.

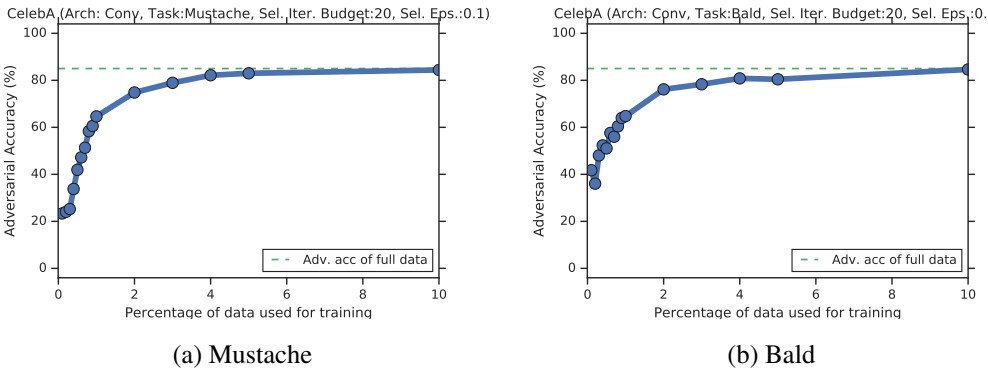

(a) Mustache                            (b) Bald

Figure 8: Simulation Results that illustrate label efficiency on CelebA. Dotted line shows downstream task adversarial accuracy obtained by using all the labelled data. Once the representation is fixed, it is feasible to achieve the same accuracy with a small fraction of data.

# D    RELATED WORK

## D.1    GENERATIVE ADVERSARIAL NETWORKS (GANS)

GANs are presented as neural sampling models of observations $x$ of form $x = f(\zeta; \eta)$ where $f$ is typically a deep neural network with parameters $\eta$, and $\zeta$ is a realization from some simple distribution $p(\mathcal{Z})$. In the context of GANs, the function $f$ is called a generator. When the dimension of $x$

is bigger than the dimension of $\zeta$, the density $p(x)$ induced by the transformation $f$ is inevitably a degenerate distribution. Since $f$ is continuous, and it is concentrated on a subset of the data space $\mathcal{X}_f \equiv \{x : \exists \zeta, x = f(\zeta; \eta)\}$. Our use of letter $f$ and parameters $\eta$ is deliberate and we will illustrate in the sequel that the generator network of a GANs is actually analogous to a smooth encoder, where the roles of the latent variables and observations are switched, but we will first review GANs.

To fit a degenerate distribution to a dataset, the GAN approach adopts a strategy where the generator is co-trained with a second neural network $d(x; w)$ with parameters $w$ with the following objective

$$\min_\theta \max_w \left\{ \mathbb{E} \left\{ \log d(x; w) \right\}_{\mathcal{D}_{\text{real}}(x)} + \mathbb{E} \left\{ \log(1 - d(g(\zeta; \theta); w)) \right\}_{p(\zeta)} \right\} \tag{25}$$

where $\mathcal{D}_{\text{real}}(x)$ is the empirical data distribution. This objective is (unfortunately) referred as an *adversarial objective* in the literature, not to be confused with adversarial attack mechanism in the context of supervised learning Madry et al. (2017). The function $d$ is called a *discriminator*. After replacing expectations with finite sample averages, this objective enforces that in a dataset that contains both synthetically generated (fake) and real examples, the classification function $d$ should increase the correct classification rate by discriminating fakes from real examples while the generator $f$ should decrease the detection rate of fake examples. When $0 \le d(\cdot) \le 1$, which is the case for a classifier, one can also write the objective as

$$\min_\theta \max_w \left\{ \mathbb{E} \left\{ l(x; w) \right\}_{\mathcal{D}_{\text{real}}} - \mathbb{E} \left\{ l(f(\zeta; \eta); w)) \right\}_{p(\zeta)} \right\} \tag{26}$$

where $l(x; w) = \log d(x; w)$. This form also highlights an interesting alternative formulation and an interpretation in terms of optimal transport. In fact, not long after the seminal work of Goodfellow et al. (2014), the mechanism beyond the GAN objective and its direct connection to the theory of optimal transport has been recognized by the seminal paper Arjovsky et al. (2017) where the problem is further framed as

$$\min_\theta \max_w \left\{ \mathbb{E} \left\{ l(x; w) \right\}_{\mathcal{D}_{\text{real}}(x)} - \mathbb{E} \left\{ l(\bar{x}; w) \right\}_{\mathcal{D}_{\text{fake}}(\bar{x}; \theta)} \right\} \tag{27}$$

with the constraint that $|l(x; w) - l(\bar{x}; w)| \le \|c(x, \bar{x})\|$, i.e. $l$ is a Lipschitz function for some $L$ where $\|c(x, \bar{x})\| \le L \|x - \bar{x}\|$. Here, $\mathcal{D}_{\text{fake}}(\bar{x}; \theta)$ is the fitted density of $\bar{x} = f(\zeta; \eta)$. This is the dual formulation of the optimal transport problem, that can be understood as an economic transaction between a customer and a shipment company. Here, the difference $l(x; w) - l(\bar{x}; w)$ can be interpreted as the profit made by the company for the shipment of one unit of mass from $x$ and to $\bar{x}$, and the Lipschitz condition ensures that it makes still sense for the customer to make use of the services of the company rather than simply doing the transport of her own (Solomon, 2018). The customer wants to pay less, so she should minimize the profit of the company. This can be achieved by changing the desired delivery distribution $\mathcal{D}_{\text{fake}}$ by adjusting $\theta$, so that the transfer from the fixed source distribution $\mathcal{D}_{\text{real}}$ is minimized. Ideally, when $\mathcal{D}_{\text{fake}} = \mathcal{D}_{\text{real}}$, there is nothing to transfer and no cost is incurred. This objective also minimizes the Wasserstein distance between the actual data distribution $\mathcal{D}_{\text{real}}$ and the fake data distribution $\mathcal{D}_{\text{fake}}$ as given by the generator.

Once the GAN objective can be viewed as minimizing a particular Wasserstein distance, it is rather straightforward to view it as a maximizer of a particular ELBO corresponding to a particular smooth encoder, albeit in one where the positions of the observations and the latents are exchanged and a very large coupling coefficient $\gamma$ is chosen. Moreover, the variational marginals have specific forms: One marginal $Q_a(X)$ is chosen as the empirical data distribution and the other marginal is chosen as having the form of a neural sampler $Q_b(X_b) = \int q(X_b | Z_b, \eta) p(Z_b) dZ_b$.

The artificial extended target becomes

$$p(Z, Z' | X, \theta) \propto \int dX_b p(Z | X, \theta) p(Z' | X_b, \theta) \psi(X, X_b) \tag{28}$$

It can be seen that the ELBO in this case becomes

$$\begin{aligned} \log p(Z, Z' | X, \theta) \ge\ & \mathbb{E} \left\{ \log p(Z | X, \theta) \right\}_{Q_a(X)} + \mathbb{E} \left\{ \log p(X) \right\}_{Q_a(X)} \\ & + \mathbb{E} \left\{ \log p(Z' | X_b, \theta) \right\}_{Q_b(X_b)} + \mathbb{E} \left\{ \log p(X_b) \right\}_{Q_b(X_b)} \\ & - \frac{\gamma}{2} \mathbb{E} \left\{ c(X_a, X_b) \right\}_{Q(X_a, X_b)} + H(Q(X_a, X_b)) \end{aligned} \tag{29}$$

Now, by taking the coupling $\gamma$ sufficiently large, the coupling term dominates the lower bound and we obtain the Wasserstein minimization objective. The random draws from $p(Z)$ become the selection mechanism. Moreover, the terms that depend on the artificial target $p(Z|X, \theta)$ become also irrelevant so in this regime the problem becomes just solving the optimal transport problem between $Q_a$ and $Q_b$.

A link between entropic GANs and VAEs is also pointed at in the literature, albeit for calculating a likelihood for GANs Balaji et al. (2018). However, our motivations as well as the interpretation of the connection is quite different and we view the GAN decoder as an instance of the smooth encoder.

### D.2 DISENTANGLED REPRESENTATIONS AND $\beta$-VAE

Targeting the encoder to an augmented distribution different than the decoder us the freedom to express some extensions of VAE in the same framework. One of such extensions is the $\beta$-VAE, quite often used for controlling representations replaces the original variational objective (1) with the following objective

$$\log p(X = x|\theta) \quad \geq \quad \mathbb{E}\left\{\log p(X = x|Z, \theta)\right\}_{q(Z|X_a=x,\eta)} - \beta D_{\mathrm{KL}}(q(Z|X_a = x, \eta)||p(Z)) \tag{30}$$

The justification in the original paper Higgins et al. (2017) is obtained from an implicit robustness criteria where $D_{\mathrm{KL}}(q(Z|X_a = x, \eta)||p(Z)) < \epsilon$ and $\beta$ appears in a Lagrangian formulation. Hoffman & Johnson (2016) have also provided an alternative justification.

In our formulation, $\beta$ can be simply interpreted as a dispersion term that is related to the number of points selected by the selection mechanism. To see this, suppose the selection mechanism chooses $\beta - 1$ points $x_{b,i}$ where $i = 1 \ldots \beta - 1$ that are identical to the true observation $x = x_{b,i} = x'_i$ for $i = 1 \ldots \beta - 1$.

$$p(X|\theta) \quad = \quad \int dX'_{1:\beta-1} p(X, X'_{1:\beta-1}|\theta) \tag{31}$$

$$\propto \quad \int dX'_{1:\beta-1} dZ dZ'_{1:\beta-1} p(X|Z, \theta) p(Z) \left(\prod_{i=1}^{\beta-1} p(X'_i|Z'_i, \theta) p(Z'_i)\right) \tag{32}$$

$$= \quad \int dZ dZ'_{1:\beta-1} p(X|Z, \theta) p(Z) \left(\prod_{i=1}^{\beta-1} p(Z'_i)\right) \tag{33}$$

Now, instead of integrating out $Z'_{1:\beta-1}$, we choose a variational distribution with identical marginals of form

$$Q(Z, Z'_{1:\beta-1}) = q(Z|X_a = x, \eta) \prod_{i=1}^{\beta-1} q(Z'_i|X_{b,i} = x, \eta) \tag{34}$$

The variational lower bound becomes identical to the $\beta$-VAE objective as

$$\log p(X|\theta) \quad \geq \quad \mathbb{E}\left\{\log p(X|Z, \theta)\right\}_{q(Z|X_a=x,\eta)} + \mathbb{E}\left\{\log p(Z)\right\}_{q(Z|X_a=x,\eta)} \tag{35}$$

$$+ \sum_{i=1}^{\beta-1} \mathbb{E}\left\{\log p(Z'_i)\right\}_{q(Z'_i|X_{b,i}=x,\eta)} + H(Q(Z, Z'_{1:\beta-1})) \tag{36}$$

$$= \quad \mathbb{E}\left\{\log p(X = x|Z, \theta)\right\}_{q(Z|X_a=x,\eta)} - \beta D_{\mathrm{KL}}(q(Z|X_a = x, \eta)||p(Z)) \tag{37}$$

where the last step follows due to the functional form of the variational distribution.

## E TECHNICAL RESULTS

### E.1 BATCH ELBO

In section 2.2, we have defined a batch ELBO (2). To see the connection to VAE ELBO (1)

$$\log p(X = x|\theta) \quad \geq \quad \mathbb{E}\left\{\log p(X = x|Z, \theta)\right\}_{q(Z|X=x,\eta)} - D_{\mathrm{KL}}(q(Z|X = x, \eta)||p(Z)) \equiv \mathcal{B}_x(\eta, \theta)$$

we first define the empirical data distribution $\pi(X) = \frac{1}{N}\sum_{i=1}^{N}\delta(X - x_i)$. We can now write

$$\log p(X = x|\theta) \geq \mathbb{E}\left\{\log p(X = x|Z,\theta)\right\}_{q(Z|X=x,\eta)} - D_{\mathrm{KL}}(q(Z|X = x,\eta)||p(Z)) \equiv \mathcal{B}_x(\eta,\theta)$$

$$
\begin{aligned}
\frac{1}{N}\sum_{i=1}^{N}\log p(X = x_i|\theta) &= \frac{1}{N}\sum_{i=1}^{N}\mathbb{E}\left\{\log p(X = x_i|Z,\theta)\right\}_{q(Z|X=x_i,\eta)} \\
&\quad -\frac{1}{N}\sum_{i=1}^{N}\mathbb{E}\left\{\log q(Z|X = x_i,\eta)\right\}_{q(Z|X=x_i,\eta)} \\
&\quad +\frac{1}{N}\sum_{i=1}^{N}\mathbb{E}\left\{\log p(Z)\right\}_{q(Z|X=x_i,\eta)} \\
&= \mathbb{E}\left\{\log p(X|Z,\theta)\right\}_{q(Z|X,\eta)\pi(X)} - \mathbb{E}\left\{\log q(Z|X,\eta)\right\}_{q(Z|X,\eta)\pi(X)} \\
&\quad +\mathbb{E}\left\{\log p(Z)\right\}_{q(Z|X,\eta)\pi(X)} \\
&\quad -\mathbb{E}\left\{\log \pi(X)\right\}_{q(Z|X,\eta)\pi(X)} + \mathbb{E}\left\{\log \pi(X)\right\}_{\pi(X)} \\
&= -D_{\mathrm{KL}}(q(Z|X,\eta)\pi(X)||p(X|Z,\theta)p(Z)) + const
\end{aligned}
$$

This result shows that the ELBO is minimizing the KL distance between one exact and one approximate factorization of the joint distribution $p(X, Z) = p(X|Z,\theta)p(Z) \approx q(Z|X,\eta)\pi(X)$.

### E.2   WHY IS THE DECODER TYPICALLY SMOOTH AFTER THE VAE TRAINING?

In the context of a VAE, the smoothness of the decoder is implicitly enforced by the highly constrained encoder distribution and the dynamics of an SGD based training. In the sequel, we will illustrate that, if two latent coordinates are sufficiently close, the decoder mean mapping is forced to be bounded.

In a standard VAE, the encoder output for each data point is conditionally Gaussian as $q(Z|X = x;\eta) = \mathcal{N}(f_\mu(x;\eta), f_\Sigma(x;\eta))$. The decoder is chosen as $p(X|Z = z;\eta) = \mathcal{N}(g(z;\theta), vI)$. The decoder parameters $\theta$ under the ELBO depend only on the data fidelity term $\|x - g(z;\theta)\|^2/v$.

For a moment, assume that the encoder is fixed and focus on a single data point $x$. During training, a set of latent state vectors $z_i$ for $i = 1 \ldots T$ are sampled from the conditionally Gaussian encoder distribution. When the dimension of the latent space $D_z$ is large, these samples $z_i$ will be with high probability on the typical set. The typical set of a nondegenerate Gaussian distribution is approximately the surface of a Mahalanobis ball, a compact hyper-ellipsoid $M(x)$ centered at $f_\mu(x;\eta)$ with scaling matrix $f_\Sigma(x;\eta)^{1/2}$.

If we assume that the training procedure is able to reduce the error in the sense that $\|x - g(z_i;\theta)\| \leq E$ for all $z_i$ where $E$ is a bound on the error magnitude for $z_i$ sampled from the encoder, the decoder is forced to give the same output for each point approximately on $M(x)$. For a point $z_a$ drawn from $q(Z|X = x;\eta)$ we have

$$\|z_a - f_\mu(x;\eta)\|_K \approx \sqrt{D_z} \quad \text{with high probability}$$

where $K = f_\Sigma(x;\eta)^{-1}$ and $\|x\|_K \equiv \sqrt{x^\top K x}$.

For a point $z_b$ independently drawn from $q(Z|X = x;\eta)$, by the triangle inequality we have

$$\|g(z_a;\theta) - g(z_b;\theta)\| \leq 2E \tag{38}$$

where the Mahalanobis distance

$$2\sqrt{D_z} \approx \|z_a - z_b\|_K \leq \frac{1}{\sqrt{\lambda_{\min}}}\|z_a - z_b\|$$

where $\lambda_{\min}$ is the smallest eigenvalue of the covariance matrix. Hence the distance is also bounded when the variance is not degenerate and minimum distance will be on the order of $\|z_a - z_b\| \approx 2\sqrt{D_z\lambda_{\min}}$ so we expect the ratio to be bounded

$$\|g(z_a;\theta) - g(z_b;\theta)\|/\|z_a - z_b\| \leq E/\sqrt{D_z\lambda_{\min}} \tag{39}$$

We see that the ELBO objective enforces the decoder to be invariant on the typical set of $q(Z|X = x; \eta)$, where most of the probability mass is concentrated.

Now, for each data point $x$, the corresponding latent space hyper-ellipsoid $M(x)$ are forced to be large in the sense of having a large determinant by the entropy term of the encoder that promotes large log-determinant. The size of $M(x)$ is also controlled by the prior fidelity term, avoiding blowing up. Hence the union $\cup_{x \in \mathcal{X}} M(x)$, where $\mathcal{X}$ is the dataset, will approximately cover the latent space when the encoder has converged and on each hyper-ellipsoid $M(x)$ the decoder will be enforced to be smooth.

### E.3   SMOOTHNESS OF THE SMOOTH ENCODER

In this section we show that the smooth encoder training forces a small Lipschitz constant for the encoder mean mapping. To simplify the argument, we will assume that the variance mapping of the encoder would be a constant function that does not vary with $x$, i.e., $f_\Sigma(x; \eta) = \Sigma(\eta)$. The latter assumption could be removed by considering a metric on the joint space of the means and covariance.

Using the adversarial selection mechanism, during training we solve the following problem using PGD:

$$x^* = \arg \max_{x' : \|x' - x\|_p \leq \epsilon} \mathcal{WD}(q(Z|X = x, \eta), q(Z|X' = x', \eta))$$

Assuming that PGD finds the global maximum at the boundary of the $\epsilon$-ball where $\|x - x^*\|_p = \epsilon$, under constant variance assumption for the encoder we can see that the Wasserstein divergence simply becomes the square distance between mean mappings

$$\mathcal{WD}(q(Z|X = x, \eta), q(Z|X' = x^*, \eta)) = \|f_\mu(x; \eta) - f_\mu(x^*; \eta)\|_2^2$$

We know that the SE ELBO objective has to minimize this distance for any coupling term $\gamma$ so the procedure actually tries to reduce the local Lipschitz constant $L(x)$ around data point $x$

$$L(x) = \frac{\|f_\mu(x; \eta) - f_\mu(x^*; \eta)\|}{\|x - x^*\|_p} \leq \frac{E}{\epsilon}$$

and promotes smoothness where $E$ is an upper bound on the change in the representation $\|f_\mu(x; \eta) - f_\mu(x^*; \eta)\| \leq E$.

