# OpenReview forum: "Adversarially Robust Representations with Smooth Encoders"
_ICLR.cc/2020/Conference — Accept (Poster)_

### Official Review · AnonReviewer3 · 2019-10-23
**Official Blind Review #3**

**Rating:** 6

**Review:**

This paper studies the vulnerability of representations learned by variational auto-encoders (VAE). It first show that the learned representation of VAE is susceptible to small changes, similar to the adversarial examples in supervised learning setting. Then propose a regularization method, called smooth encoder, to improve the robustness of the representation. Experiments are conducted on several benchmark datasets to show the effectiveness of the method.

Overall I find the idea interesting and the experimental results promising. The following are my detailed comments.

a About the theory
The illustration of the problem in VAE is interesting. However, one missing point is to theoretically quantify the effect of the proposed regularization (in some simple cases). In particular, it is claimed that the regularization could make the encoder smoother and the experimental results clearly justifies it. What would be better is to show in which sense/measure the encoder is smoother and provide some theoretical guarantee about it. (for instance smaller Lipschitz constant?)

b About the Experiment
The experimental section is clear and promising. I just have one question about the evaluation on the robustness of the VAE representation. In particular, a linear classifier is concatenated right after the VAE representation and it is not clear to me where it is concatenated. Is it right after the layer of \mu and \Sigma or in later layers? If it is in the later layers, the VAE is outputting a distribution, then how does the accuracy measured?

Minor comment:
I think it is unnecessary to introduce the new term selection strategy because it is just an adversarial training with respect to a different loss. In particular, the loss is the Wasserstein distance between the latent space vectors instead of a supervised loss. For simplicity, it could be just named as latent space adversarial training. (this is just a suggestion, which will not change my decision)

**Experience Assessment:**

I have read many papers in this area.

**Review Assessment: Checking Correctness Of Derivations And Theory:**

I assessed the sensibility of the derivations and theory.

**Review Assessment: Checking Correctness Of Experiments:**

I assessed the sensibility of the experiments.

**Review Assessment: Thoroughness In Paper Reading:**

I read the paper thoroughly.

---

> ### Author Response · Authors · 2019-11-13
> **Responses to the comments by Rev#3**
>
> We thank the reviewer for the constructive feedback.
>
> ++++++++++++++++++++++
> #3a (The illustration of the problem in VAE is interesting. However, one missing point is to theoretically quantify the effect of the proposed regularization (in some simple cases). In particular, it is claimed that the regularization could make the encoder smoother and the experimental results clearly justifies it.
> What would be better is to show in which sense/measure the encoder is smoother and provide some theoretical guarantee about it. (for instance smaller Lipschitz constant?)
> ..........
>
> We agree with the reviewer and we provide a more formal argument in appendix E.3.
>
> Under the assumption of constant encoder variance, the selection mechanism finds a point in the vicinity of the original data point that maximally changes the distance between the means of latent representations. The SE ELBO in this case forces the means to be close hence in effect promoting a reduction in the corresponding local Lipschitz constant of the encoders mean mapping.
>
> ++++++++++++++++++++++
> #3b) (I just have one question about the evaluation on the robustness of the VAE representation. In particular, a linear classifier is concatenated right after the VAE representation and it is not clear to me where it is concatenated. Is it right after the layer of \mu and \Sigma or in later layers? If it is in the later layers, the VAE is outputting a distribution, then how does the accuracy measured?)
> ..........
>
> We adopt a two step experimental protocol, where we first train encoder-decoder pairs agnostic to any downstream task. Then we fix the representation, that is we freeze the encoder parameters and only use the deterministic mean mapping of the encoder as the latent representation, then train a simple linear classifier based only on the encoder means using standard techniques.
>
> We have updated the first paragraph of the experimental section as above.
>
> In all the experiments the results are based on the means, where the linear classifier is concatenated just right after the final encoder layer, by omitting the variances. The adversarial accuracy reported for the accuracy of the linear layer attached to the encoder. The smoothness is only enforced at the output of the encoder. The decoder, or representations emerging at other layers than the encoder outputs are never used in our evaluation.
>
> ++++++++++++++++++++++
> #3c) (think it is unnecessary to introduce the new term selection strategy because it is just an adversarial training with respect to a different loss. In particular, the loss is the Wasserstein distance between the latent space vectors instead of a supervised loss. For simplicity, it could be just named as latent space adversarial training. (this is just a suggestion, which will not change my decision)
> ..........
>
> The term ‘latent space adversarial training’ is certainly an accurate description of the approach and we will consider it. The reason we have used the term selection mechanism is twofold:
> *	Adversarial training in the context of unsupervised learning is an overloaded term with connotation to GAN’s or adversarial objectives.
>
> *	The selection mechanism can be quite general beyond adversarial attacks such as small perturbations constrained in an \epsilon-norm-ball. For example, in domains such as image recognition, we indeed wish to generate examples that are semantically related to the original data point. While we don’t investigate these extensions in the present paper, we prefer to use a more general term as a 'selection mechanism'.

---

### Official Review · AnonReviewer2 · 2019-10-26
**Official Blind Review #2**

**Rating:** 3

**Review:**

This paper analyzes the shortcoming of VAE objective, and propose a regularization method based on a selection mechanism that creates a fictive data point by explicitly perturbing an observed true data point. It is lead to Wasserstein distance between representations. Experiments are made on three datasets; ColorMNIST, MNIST, and CelebA, which shows superior performance on adversarial accuracy while similar accuracy to VAE on nominal accuracy.
The paper is well-organized and well-written. The point is clear and the proposed algorithm is valid. The only problem of the paper is the improvement on the experiment is marginal. Although adversarial accuracy is far better (like 0% vs 50%), it is apparent that the vanilla VAE is fragile to the adversarial examples because the added noise is intended so. Thus I can not say this is a fair comparison and because the superiority of the proposed algorithm is shown in only this point, I am not sure the proposed algorithm is surely useful.


**Experience Assessment:**

I have published one or two papers in this area.

**Review Assessment: Checking Correctness Of Derivations And Theory:**

I assessed the sensibility of the derivations and theory.

**Review Assessment: Checking Correctness Of Experiments:**

I assessed the sensibility of the experiments.

**Review Assessment: Thoroughness In Paper Reading:**

I read the paper at least twice and used my best judgement in assessing the paper.

---

> ### Author Response · Authors · 2019-11-13
> **Response to the comment by Rev#2**
>
> We thank the reviewer for their time to review the paper.
>
> +++++++++++++
> #2a (The only problem of the paper is the improvement on the experiment is marginal. Although adversarial accuracy is far better (like 0% vs 50%), it is apparent that the vanilla VAE is fragile to the adversarial examples because the added noise is intended so. Thus I can not say this is a fair comparison and because the superiority of the proposed algorithm is shown in only this point, I am not sure the proposed algorithm is surely useful.)
> ....................
>
>
> Representation learning is increasingly used as a paradigm in large scale AI systems, where a representation is trained on large amounts of unlabeled data and then used for several downstream prediction tasks. In this context, if it happens that a representation is overly sensitive to small changes in the input, it can have catastrophic consequences for the downstream tasks the representation is used for. Thus, we believe that the adversarial accuracy on downstream tasks, which measures the performance of the model under imperceptible perturbations of the input, is a valuable metric by which to judge a representation learning method. Since VAEs are a popular paradigm in representation learning, we focused on adversarial accuracy of representations learned by VAEs in this work.
> We would welcome any other concrete suggestion to make a more fair comparison.

---

### Official Review · AnonReviewer1 · 2019-10-31
**Official Blind Review #1**

**Rating:** 8

**Review:**

This is a very interesting paper, I believe, a solid contribution to Variational Autoencoders. The basic argument is that encoders in VAEs are highly susceptible to noise in input data, whereas decoders are not. This argument is supported with a full fledged section 2.2, reformulating ELBO objective of VAEs, and introducing a VAE with discrete latent variables and discrete observations, so as to easily understand why and where VAEs fail.

To make encoders robust to noise in inputs, it is proposed to generate new fictive data points in the neighborhood of original data points so as to ensure that the latent representations of a data point and its fictive version are similar in "some sense" as part of the proposed regularization term. The implementation of this idea is solid in the paper, relating it to theoretical concepts such as  "entropy regularized entropy transport problem", "Wasserstein distance", etc. The most important point is that, it is easy to extend an encoder of an existing VAE with the proposed algorithm, while letting a decoder be untouched as the latter is shown to be robust/smooth anyways (in sec 2.2). It is also discussed on how to generate fictive samples, including but not restricted to approaches like projected gradient descent based adversarial attacks.

Section 2.2 can be improved further, in terms of presentation. This is the most important section which can be of interest to the community to understand VAEs' limitations, a good contribution on its own. Though challenging, I encourage the authors to improve the exposition in this section as much as possible.

Introduction is written beautifully. Good job, done!

For instance, some explanation about variables, m_j, u_i, their distribution.

How do you relate the Eq. 1 with the standard ELBO. (some reference to derivation?)

Is it not possible to explain limitations of present VAEs without introducing the particular von Mise like parameterization (last equation of page 3). I am not suggesting that you should remove it. The connections between the two could be more explicit, though I understand that it is already mentioned in the paper, "parameterization emulates a high capacity network that can model any functional relationship between latent states and observations...".

In this context, I found the explanation after Eq. 2 to be intuitive in regards to inefficiency of encoders. If I understand correctly, to put it in even simpler terms, the encoding neural network is overfitting mapping from input data points to the latent representations, not performing any learning for the unseen data points at all; on the other hand, decoder explores the space of latent variables well because it is modeled as a Gaussian?

Some of the new equations should be numbered for easy reference.

On page 4, the flow is a bit abrupt. Right after Fig. 3, there are points 1 and 2 added without any note on what these two points (items in latex) are about.

I found point 1 very confusing in page 4. On the other hand, point 2 is beautifully written. Though, it could be made explicit in the latter on why encoders found in VAE are not smooth, referring to Fig 2, 3.

There are minor grammar mistakes making some of sentences incoherent or confusing, in the paper. Something to do with style of language. I think, overall, language can be improved. Though, technical flow of the paper is great, and introduction is written very well, pointing out very important bold insights about the literature on unsupervised representation learning. I would say, it is a very well written paper, which is an enjoyable read, despite some of the grammar mistakes which can be easily fixed by proof reading.

Experimental evaluation is sufficient.

Last but not the least, one could argue that we are going to the literature of kernel function based methods, or markov random fields, to improve the neural network models. This is a general trend we are observing. It is interesting to see new models such as the proposed one, getting the best from both worlds. It may be worthwhile to point  out something along these lines in the paper so that other works like this can be accomplished which are bold, and advance representation learning, digging mathematical concepts from diverse domains. If I am mistaken, please feel free to point out. It is not going to be change the review. I am inspired from this work.

One practical challenge is to generate fictive data points which are not very near to existing data points. I am not sure if GANs can achieve that, either. Having such points is critical to deal with more structured noise. Any comments on this?

**Experience Assessment:**

I have published in this field for several years.

**Review Assessment: Checking Correctness Of Derivations And Theory:**

I carefully checked the derivations and theory.

**Review Assessment: Checking Correctness Of Experiments:**

I carefully checked the experiments.

**Review Assessment: Thoroughness In Paper Reading:**

I read the paper at least twice and used my best judgement in assessing the paper.

---

> ### Author Response · Authors · 2019-11-13
> **Responses to the comments by Rev#1**
>
> We thank the reviewer for their kind remarks and constructive feedback.
>
> +++++++++++++++++++++++++++++
> #1a) Improve and clarify section 2.2 [the example illustrating the problem with the VAE objective]: Alternative explanation for limitations of VAE additional to the von-Mises example
> .............................
>
> We believe that the von-Mises example serves as an illustrative example where the densities can be computed precisely and visualized, so that the problem of non-smoothness in the decoder can be easily understood. We have rewritten this section and clarified the explanation of non-smoothness further in the revised draft. We have added a short clarification to the appendix E.1. See also response #1c).
>
> +++++++++++++++++++++++++++++
> #1b) Clarification of the relation between the Batch ELBO and the standard ELBO. (reference to derivation?)
> .............................
>
> The Batch ELBO can be derived by considering the target p(X|Z) p(Z) and proposing an approximation of form \hat{p}(X) q(Z|X).
>
> Starting from the original VAE ELBO that applies to a single data point, the batch ELBO will contain a sum over all data points. Viewing \pi(X) as an empirical distribution,and writing the average Batch log likelihood as an additional expectation wrt to the empirical distribution of the original ELBO will lead to (1). We have added a short clarification to the appendix E.2.
>
>
> +++++++++++++++++++++++++++++
> #1c) Further justification and improving the presentation by avoiding abrupt changes and further justification of points 1 [remarks about the smoothness of the decoder] and 2 [remark on non-smoothness of the encoder]: clarification of point 1, improving point 2 [non-smooth encoders] establishing connection.
> #1 (In this context, I found the explanation after Eq. 2 to be intuitive in regards to inefficiency of encoders. If I understand correctly, to put it in even simpler terms, the encoding neural network is overfitting mapping from input data points to the latent representations, not performing any learning for the unseen data points at all; on the other hand,  decoder explores the space of latent variables well because it is modeled as a Gaussian?)
> .............................
>
> We observe that at the end of the training, the mean mapping of the decoder, from latent space to observables, is approximately a smooth function empricially - the same observation was also made in previous papers (Kingma and Welling 2013, Rezende and Mohamed 2014).
>
> The smoothness of the decoder is implicitly enforced by the conditionally Gaussian (unimodal) encoder distribution and the nature of the ELBO loss.
> The intuition in a nutshell is the following: After the encoder converges, the latent state vectors will be approximately on a fixed ellipsoid (the typical set of a Gaussian) in the latent space as given by the encoder mapping. The ELBO objective will force the decoder mapping to be invariant on this ellipsoid.
>
> We have added an appendix section (E.2) to the new version of the paper that provides the above description more formally and in more detail.
>
> +++++++++++++++++++++++++++++
> #1d) Comments about the trends literature of kernel function based methods, or markov random fields, to improve the neural network models.
> .............................
>
> We agree with the reviewers observation and we observe the same trend. In several applications, certain specifications or domain specific prior knowledge need to be incorporated into highly flexible neural network models and indeed our approach can be viewed as a particular proposal in this direction.
>
> +++++++++++++++++++++++++++++
> #1e) (One practical challenge is to generate fictive data points which are not very near to existing data points. I am not sure if GANs can achieve that, either. Having such points is critical to deal with more structured noise. Any comments on this? )
> .............................
>
> As the reviewer has correctly identified, the selection mechanism can be quite general beyond small perturbations constrained in an \epsilon-norm-ball. In domains such as image recognition, we indeed wish to generate examples that are semantically related to the original data point. There are many different ways of accomplishing this, for example by differential renderers, black box generative models, or decoders trained GANs. To support these, our general methodology would clearly need to be extended but we believe that the pairwise coupling idea can be adopted to these situations.

---

### Public Comment · ~YiBin_Wang2 · 2023-04-21
**About hyperparameters of experiments**

I cannot obtain the corresponding $\gamma$ values for each experiment from Table 2,  and I have not found the corresponding source code for the experiment either.I would appreciate it if you could provide a detailed and specific hyperparameter setting for different experiments.

---

### Decision · Program_Chairs · 2019-12-19

**Decision:**

Accept (Poster)

**Comment:**

This paper proposes an novel way of expanding our VAE toolkit by tying it to adversarial robustness. It should be thus of interest to the respective communities.